# Body Mass Index and Antibody Persistence after Measles, Mumps, Rubella and Hepatitis B Vaccinations

**DOI:** 10.3390/vaccines10071152

**Published:** 2022-07-20

**Authors:** Marco Fonzo, Annamaria Nicolli, Stefano Maso, Lorenzo Carrer, Andrea Trevisan, Chiara Bertoncello

**Affiliations:** Department of Cardiac Thoracic Vascular Sciences and Public Health, University of Padova, Via Giustiniani 2, 35128 Padova, Italy; marco.fonzo@unipd.it (M.F.); annamaria.nicolli@unipd.it (A.N.); stefano.maso@unipd.it (S.M.); lorenzo.carrer.1@studenti.unipd.it (L.C.)

**Keywords:** body mass index, vaccines, immune response, hepatitis B vaccine, MMR vaccine

## Abstract

Overweight and obesity may cause a reduced response to vaccination. The purpose of the present research was to study the relationship between current body mass index (BMI) and antibody persistence after vaccination against measles, mumps, and rubella (MMR) and hepatitis B virus (HBV) given during childhood, as per the current vaccination schedule. The study was conducted on 2185 students at the School of Medicine, University of Padua, Italy. The mean age of the participants was 20.3 years. After adjusting for sex, age at first dose of vaccine administered, age at last dose, and age at study enrollment, no significant association was found between lack of serologic protection and BMI for either the HBV vaccine or each component of the MMR vaccine. For the first time, the absence of this relationship was demonstrated for the MMR vaccine. Given the evidence currently available, further research on BMI and vaccines in general remains desirable.

## 1. Introduction

According to the World Health Organization (WHO) estimates, in 2016, 1.9 billion adults (39%) were overweight and 650 million (13%) were obese [1].

Overweight and obesity, in addition to being the cause or concomitant cause of numerous diseases—especially of the cardiovascular system—such as dyslipidemia, type 2 diabetes, hypertension, and sleep disorders [2,3], can influence the immune response to certain infections and vaccinations and alter the efficacy of antimicrobial drugs [4,5]; overall, the immune system is negatively affected by obesity [6]. For instance, it has been shown that IgG production in response to infection with herpes simplex virus 1 and 2, enterovirus, or *Chlamydia pneumoniae* is strongly associated with fat mass [7] and that overweight and obesity are associated with higher mortality from infectious diseases in adolescents [8]. On the other hand, an association—albeit less strong—between obesity in early adulthood and prenatal/infantile infections has been observed [9], assuming that the ‘immune consequences’ due to obesity could explain the lower response to vaccines in overweight and obese individuals [6]. However, the current evidence in this regard is not unequivocal. In addition to obesity, a number of other factors such as smoking, age over 50 years, male gender, immunosuppression and low birth weight (<1 kg) have been questioned as factors reducing the response to vaccination—particularly hepatitis B vaccination [10,11,12,13].

The aim of the present study was to evaluate the effect of body mass index (BMI) in young adults on antibody persistence after childhood vaccination against hepatitis B virus (HBV) and measles, mumps, and rubella (MMR).

## 2. Materials and Methods

### 2.1. Inclusion Criteria

The following inclusion criteria were considered: (1) being born in Italy after 1991, in order to have a homogeneous cohort for the HBV vaccine with regard to age of administration [14,15]; (2) having undergone a medical examination with measurement of weight and height in order to allow a reliable calculation of BMI; (3) possessing a valid vaccination certificate issued by the Public Health Office; (4) having received three doses of HBV vaccine in the first year of life and two doses of MMR vaccine (the first in the second year of life and the second between the ages of 5 and 12 years) as required by regulations at the time of vaccination [16,17]; and (5) having no previous HBV infection.

The BMI classes are those established by the WHO [1]. All subjects with a BMI lower than 18.49 kg/m^2^ were included in the ‘underweight class’, while the three obesity classes (BMI higher than 30 kg/m^2^) were considered together.

### 2.2. Population

A total of 2185 students attending the School of Medicine, University of Padua (northeastern Italy), including 815 males and 1370 females (0.59 ratio) matriculated from 2010 to 2020, were recruited. The characteristics of the subjects (age at recruitment, age at first dose of HBV and MMR vaccine, age at second dose of MMR vaccine, and time since last dose of both vaccines) are summarized in Table 1.

### 2.3. Antibody Measurements

Anti-HBs antibodies were measured with a commercial chemiluminescent micro particle immunoassay (CMIA) until 2017 and then with a chemiluminescent immunoassay (CLIA) named LIAISON^®^ anti-HBs plus by Sorin (Saluggia, Italy). To measure MMR IgG antibodies, a commercial enzyme-linked immunosorbent assay (EIA) Enzygnost (Dade Behring, Marburg, Germany) was used. According to the recommendations of the Centers for Disease Control and Prevention (CDC), equivocal results were treated as negative [18]. The absence of serological protection was defined according to the manufacturer as follows: antibody titer for hepatitis B < 10 IU/L; for measles 350 IU/mL; for mumps antibody measurement was qualitative; and for rubella lower than 10 IU/mL.

### 2.4. Statistics and Participants’ Informed Consent

Descriptive analyses were performed using absolute and relative frequencies. Mean values and relative standard deviation were reported for continuous variables. Depending on the nature of each variable analyzed, χ^2^ and *t*-tests were conducted to compare the different groups on baseline characteristics. The effect of BMI on the risk of presenting at recruitment with an antibody titer below the suggested threshold for each of the antigens studied was assessed with a single-step binary logistic regression analysis adjusting for sex, BMI (handled as a categorical variable), age at first vaccine dose, age at last vaccine dose, and age at recruitment. Statistical significance for all tests was set at *p* ≤ 0.05 (two-sided) and confidence intervals (CIs) at 95%. Statistical analyses were performed using IBM^®^ SPSS Statistics^®^ version 23. The research was based on data collected during health surveillance, so no evaluation by an ethics committee was required. However, all subjects undergoing health surveillance signed a privacy document allowing the processing and publication of anonymous data. Data collection was conducted in accordance with the principles of the Declaration of Helsinki, in compliance with applicable national legislation and with respect for the protection of personal data.

## 3. Results

As shown in Table 2, the current serological status was not significantly associated with either BMI or sex (for all distributions investigated *p* > 0.05). However, the proportion of unprotected individuals differed substantially between the four vaccines in question—being highest for the hepatitis B vaccine and lowest for the rubella vaccine.

The above findings were confirmed by the results of the logistic regression analysis (Table 3). After adjusting for sex, age at first dose of vaccine administered, age at last dose, and age at the study recruitment, no significant association was found between the lack of serological protection and BMI.

## 4. Discussion

The influence of BMI on the effectiveness of vaccinations has long been a cause for investigation; most studies concern the hepatitis B vaccine. The rather large literature relating to the immune response to the hepatitis B vaccine and BMI is almost solely oriented on the existence of this relationship [5,19,20,21,22,23,24,25,26,27,28,29,30,31,32,33,34,35,36], while critical voices are rather isolated [37,38]. A recent study has indicated the involvement of leptin as crucial in the immunogenicity of the HBV vaccine [33]. In any case, two publications of some interest argue that by using longer needles in obese or overweight adolescents during administration, the problem of reduced or absent response to the vaccine can be solved [39,40].

As for other vaccines, there are less clear-cut positions. For example, opinions on influenza vaccine response are both that BMI influences it [41,42,43,44] and that it does not [45,46]. The same applies for the hepatitis A vaccine, with studies both favoring [47,48] and opposing [49] a relationship between BMI and immune response to the vaccine. A few studies state that the rabies vaccine shows a reduced antibody response in individuals with a BMI greater than 25 kg/m^2^ after two years [50], and that obesity is a major factor in the reduced antibody response to the papilloma virus vaccine [51]. Finally, the reduced antibody response to the tetanus vaccine in obese subjects could also depend on mechanical factors in relation to reduced absorption at the inoculum site [52], somewhat like needle length for the HBV vaccine [39,40]. The relationship between BMI and immune response after vaccination was also examined for the new SARS-CoV-2 vaccine. A lower immune response was observed after the first dose [53,54], but not after the second [55]. No differences are observed after COVID-19 infection either [56]. Finally, no studies relating BMI to immune response in MMR vaccines have been reported in the literature.

With our research, we wanted to evaluate the relationship between BMI and the serological status and antibody persistence after HBV and MMR vaccines in a homogeneous cohort: all participants were vaccinated at three months of age (first dose) against HBV (with completion of the cycle within one year of age) and against MMR between the first and second year of age (first dose) and between 5 and 12 years of age (second dose). 

Our results show that there is no relationship between BMI and the persistence of the immune response after HBV and MMR vaccines. In addition, no significant sex-related difference was found.

The main strengths of our study are a large number of cases (more than 2000 subjects) and a rigorous selection of study participants. For example, most of the research in this area has focused on the relationship between BMI and responses to the HBV vaccine and has been conducted on subjects vaccinated as adolescents or in adulthood, with the oldest participants likely to have received the plasma-derived vaccine. On the other hand, a limitation of our study is that we neither know the BMI of each subject at the time of vaccination in childhood, nor do we have documentation of any significant changes in BMI over the course of life. This limitation, however, may appear circumscribed to the extent that those who are overweight or obese in adulthood often were so in childhood. As reported in a recent systematic review conducted by Simmonds and colleagues, obese children and adolescents were about five times more likely to be obese in adulthood than those who were not. In addition, about 55% of obese children become obese in adolescence, while 80% of obese adolescents will still be obese in adulthood and about 70% will continue to be obese beyond the age of 30 [57]. Although the type of vaccine used is unknown, the inclusion criteria adopted are able to ensure homogeneity in the dose administered. Furthermore, no data were collected on other factors and behaviors that may negatively influence the immunogenicity of the vaccine, including smoking habits or concomitant diseases [32,58]. 

Although the antibody protection thresholds used in this study are internationally agreed upon, it must be considered that subjects with anti-HBs titer < 10 IU/l commonly have a prompt response to the booster dose, demonstrating a strong immunological memory [59,60].

## 5. Conclusions

By resorting to sharply defined inclusion criteria in our study, it was possible to clearly define the absence of a relationship between antibody persistence after HBV vaccine and BMI; for the first time, the absence of this relationship was demonstrated for the MMR vaccine. In any case, this remains an issue to be studied carefully, considering that the current evidence—apart from the considerable agreement on the influence of BMI on HBV vaccine response—is not unambiguous on the relationship between BMI and other vaccines.

## Figures and Tables

**Table 1 vaccines-10-01152-t001:** Characteristics of the study population by sex.

	Males	Females	All
	(*n* = 815)	(*n* = 1370)	(*n* = 2185)
Age at recruitment (years ± SD)	20.4 ± 0.9	20.2 ± 0.9	20.3 ± 0.9
Age 1st HBV vaccine (days ± SD)	88.7 ± 20.2	89.1 ± 18.9	89.0 ± 19.4
Time between HBV vaccine (last dose) and analysis (years ± SD)	19.4 ± 0.9	19.2 ± 0.9	19.3 ± 0.9
Age 1st dose MMR vaccine (years ± SD)	1.4 ± 0.2	1.4 ± 0.1	1.4 ± 0.2
Age 2nd dose MMR vaccine (years ± SD)	8.5 ± 2.1	8.3 ± 2.1	8.4 ± 2.1
Time between MMR vaccine (last dose) and analysis (years ± SD)	11.9 ± 1.9	11.9 ± 2.0	11.9 ± 2.0
Weight (kg ± SD)	73.0 ± 9.8	58.3 ± 8.5	63.8 ± 11.4
Height (cm ± SD)	179.9 ± 6.5	166.6 ± 6.0	171.6 ± 9.0
BMI (kg/m^2^ ± SD)	22.5 ± 2.6	21.0 ± 2.7	21.6 ± 2.7
Underweight * (%)	31 (3.8)	208 (15.2)	239 (10.9)
Normal weight (%)	664 (81.5)	1072 (78.2)	1736 (79.5)
Overweight (%)	113 (13.9)	78 (5.7)	191 (8.7)
Obesity ** (%)	7 (0.9)	12 (0.9)	19 (0.9)

* All subjects with a BMI lower than 18.49 kg/m^2^; ** all together the three classes of obesity.

**Table 2 vaccines-10-01152-t002:** Distribution of the serological lack of protection by sex and BMI.

	Measles			Mumps			Rubella			Hepatitis		
	*n*	(%)	*p*	*n*	(%)	*p*	*n*	(%)	*p*	*n*	(%)	*p*
Males	219	26.9	0.195	110	13.5	0.525	44	5.4	0.051	407	49.9	0.471
Females	334	24.4		172	12.6		50	3.7		706	51.5	
Normal weight	445	25.6	0.545	225	13.0	0.626	77	4.4	0.134	894	51.5	0.485
Underweight *	54	22.6		35	14.6		5	2.1		120	50.2	
Overweight	51	26.7		20	10.5		12	6.3		88	46.1	
Obesity **	3	15.8		2	10.5		0	0.0		11	57.9	

* All subjects with a BMI lower than 18.49 kg/m^2^; ** all together the three classes of obesity.

**Table 3 vaccines-10-01152-t003:** Logistic regression analysis. Outcome investigated: serological lack of protection.

	Measles			Mumps			Rubella			Hepatitis		
	AOR	(95% CI)		AOR	(95% CI)		AOR	(95% CI)		AOR	(95% CI)	
Male sex	1.09	0.88	1.33	1.15	0.88	1.49	1.39	0.91	2.14	0.97	0.81	1.16
BMI (ref. Normal weight)												
Underweight *	0.86	0.62	1.20	1.20	0.81	1.77	0.49	0.20	1.25	0.92	0.70	1.21
Overweight	1.07	0.76	1.50	0.78	0.48	1.28	1.31	0.69	2.49	0.81	0.60	1.10
Obesity **	0.59	0.17	2.04	0.79	0.18	3.47	-	-	-	1.26	0.50	3.16

* All subjects with a BMI lower than 18.49 kg/m^2^; ** all three classes of obesity together; AOR: adjusted odds ratio; 95% CI: 95% confidence interval.

## Data Availability

Raw data are available on request from the corresponding author.

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
