# Peer review of "Body Mass Index and Antibody Persistence after Measles, Mumps, Rubella and Hepatitis B Vaccinations"

_vaccines, 2022, doi:10.3390/vaccines10071152_

Round 1
Reviewer 1 Report
This reviewer has the following concerns about this manuscript.
1) Is this paper a brief report since the length and results are not long enough.
2) The introduction should be extended to clearly explain the motivation and object of this study.
3) The results are vague and must be explained more. Is there any relationship between the parameters authors investigated?
Author Response
Reviewer 1
1) Is this paper a brief report since the length and results are not long enough.
Reply: As authors, we defer to the editor for considerations on these aspects.
2) The introduction should be extended to clearly explain the motivation and object of this study.
Reply: Thank you for your suggestion. In this regard, the introductory section was expanded, as was the statement on the objective of the study.
3) The results are vague and must be explained more. Is there any relationship between the parameters authors investigated?
Reply: As stated both in the abstract and in the body of the manuscript, we investigated whether there was a relationship between the variables investigated. Since the current evidence is not unambiguous, our study aims to increase the available evidence.
Reviewer 2 Report
This paper examined antibody persistence from MMR and Hepatitis B vaccination in medical students.
The study did not mention institutional review board approval. "Unprotected" is not defined, i.e. noting the cut-offs for each vaccine antibody level should be recorded (e.g.>=10mIU/mL for anti-HBs) to help define the term "unprotected".
Smoking, as well as obesity, can affect anti-HBs responses, mention should be made if this was examined or overlooked.
Many individuals with levels of anti-HBs (<10mIU/mL) are protected by the anamnestic response to HBV infection, the authors should add a comment in this regard in the Discussion.
1. What is the main question addressed by the research?
The authors state in the abstract (and methodology)…The purpose of the present research was to study the relationship between current body mass index (BMI) and antibody persistence after vaccination against measles, mumps, and rubella (MMR) and hepatitis B virus (HBV) given during childhood, as per the current vaccination schedule.
2. Do you consider the topic original or relevant in the field, and if so, why?
The topic is not original, however, revisiting this is worthy, given the worldwide and growing problem of obesity. However, the problem with the paper is that there were very few obese participants in the study (looking at the figures… the were 19 obese of whom 11 had HBV titres <10miu/mL, versus 1736 normal weight individuals of whom 1113 were non-immune, so it is not surprising a Chi-square gives a p value of 0.78. So I think the conclusions made are not justified.
3. What does it add to the subject area compared with other published material? The issue remains controversial; the authors have pointed out publications indicating conflicting results. Their article does not help sort this out because they had too few obese participants to provide a valid result. One must question why 64% of normal-weight individuals had nonprotective levels of anti-HBs. What proportion were smokers? The study did not control for smoking. Why not?
4. What specific improvements could the authors consider regarding the methodology?
I think there are insufficient numbers of obese participants to draw conclusions. They need to refashion the paper (see 7 below).
5. Are the conclusions consistent with the evidence and arguments presented and do they address the main question posed?
I think there are insufficient numbers of obese participants to draw conclusions. Therefore, I think the paper should be rejected.
6. Are the references appropriate? Probably OK. Now irrelevant!
7. Please include any additional comments on the tables and figures.
The authors could present the data as scattergrams (body weight versus antibody titre). This could provide a useful angle, and provide a means of publishing a different and meaningful paper.
Author Response
Reviewer 2
The study did not mention institutional review board approval.
Reply: Thank you for your suggestion. A statement in this regard was added also in the body of the manuscript, while before it was reported only as a paragraph after conclusions.
"Unprotected" is not defined, i.e. noting the cut-offs for each vaccine antibody level should be recorded (e.g.>=10mIU/mL for anti-HBs) to help define the term "unprotected".
Reply: We definitely agree with you. It was a clear oversight on our part. A paragraph was added.
Smoking, as well as obesity, can affect anti-HBs responses, mention should be made if this was examined or overlooked.
Reply: Unfortunately, data on neither the participants' smoking habits nor concomitant diseases were available, so we indicated this as a limitation of our study.
Many individuals with levels of anti-HBs (<10mIU/mL) are protected by the anamnestic response to HBV infection, the authors should add a comment in this regard in the Discussion.
Reply: We agree with this and we added a short paragraph on this topic at the end of the discussion.
- What is the main question addressed by the research?
The authors state in the abstract (and methodology)…The purpose of the present research was to study the relationship between current body mass index (BMI) and antibody persistence after vaccination against measles, mumps, and rubella (MMR) and hepatitis B virus (HBV) given during childhood, as per the current vaccination schedule.
Reply: The text has been modified accordingly.
- Do you consider the topic original or relevant in the field, and if so, why?
The topic is not original, however, revisiting this is worthy, given the worldwide and growing problem of obesity. However, the problem with the paper is that there were very few obese participants in the study (looking at the figures… the were 19 obese of whom 11 had HBV titer <10miu/mL, versus 1736 normal weight individuals of whom 1113 were non-immune, so it is not surprising a Chi-square gives a p value of 0.78. So I think the conclusions made are not justified.
Reply: In our humble opinion, the overall sample size is adequate and the calculation of the 95%CI of the adjusted ORs allows us to draw the following conclusions. In light of this, we consider the entire analysis to be robust and the "conclusion appears to be valid" as also stated by Reviewer n.3.
- What does it add to the subject area compared with other published material? The issue remains controversial; the authors have pointed out publications indicating conflicting results. Their article does not help sort this out because they had too few obese participants to provide a valid result. One must question why 64% of normal-weight individuals had nonprotective levels of anti-HBs. What proportion were smokers? The study did not control for smoking. Why not?
Reply: See previous comments regarding both the number of participants and the percentage of smokers in our sample.
- What specific improvements could the authors consider regarding the methodology?
I think there are insufficient numbers of obese participants to draw conclusions. They need to refashion the paper (see 7 below).
Reply: See answer to comment no. 2.
- Are the conclusions consistent with the evidence and arguments presented and do they address the main question posed?
I think there are insufficient numbers of obese participants to draw conclusions. Therefore, I think the paper should be rejected.
Reply: See answer to comment no. 2 and 4.
- Are the references appropriate? Probably OK. Now irrelevant!
Reply: Ok
- Please include any additional comments on the tables and figures.
The authors could present the data as scattergrams (body weight versus antibody titre). This could provide a useful angle, and provide a means of publishing a different and meaningful paper.
Reply: As the entire paper focuses on the lack of serological protection, we have treated this variable not as a continuous variable, but rather as a dichotomous one in order to better answer the scientific question we wanted to address. As for the meaningfulness of the article, we defer to the editor.
Reviewer 3 Report
Overweight and obesity are known to have many adverse effects on many systems within the body including the immune and endocrine systems. While the effects of obesity and overweight are causally related to the occurrence of numerous diseases there appears to be far less known about the effects of overweight and obesity on the immune response and persistence of antibodies post-vaccination against hepatitis B virus (HBV), measles, mumps and rubella (MMR). This study sought to address this lack of data by investigating the relationship between current body mass index (BMI) in young adults and the persistence of antibodies in young adults that received HBV and MMR vaccination during childhood.
For a study of this type the inclusion criteria and subject numbers are clearly of critical importance. The authors appear to have carefully chosen the study subjects with adequate attention being paid to the timing of their vaccine doses of HBV and MMR. This is adequately described in 2.1 Inclusion criteria.
The study cohort of 2185 students were recruited consisting of 815 males and 1,370 females achieving a comparable standard of education between 2010 and 2020. The data presented in Table 1 effectively summarize the relevant details of the study population including details of BMI and overweight and underweight %.
Antibody measurements were carried out using commercially available kits.
Statistical analysis of data was carried out using valid statistical techniques and these are briefly described in section 2.4.
In a brief but adequate discussion the authors critically evaluate the results of this study by considering previous relevant and related studies. The authors point out that the sharply defined inclusion criteria included in their study enabled them to reliably conclude that there was no significant relationship between antibody persistence after HBV vaccine and BMI. This conclusion appears to be valid. Furthermore, in a new finding, the absence of a relationship was demonstrated between antibody persistence after MMR vaccine and BMI. Many readers might have the impression that BMI would influence antibody production even by physically influencing the delivery of vaccines into different tissue types. In dispelling this idea the study makes a significant contribution to current knowledge.
The assumption is made that the BMI of participants did not change significantly between childhood and adulthood and this is largely based upon the significant review of Simmonds et al (2016). Overall the literature cited in the discussion is thorough and it is generally quite relevant to this paper.
Author Response
Reviewer 3
Overweight and obesity are known to have many adverse effects on many systems within the body including the immune and endocrine systems. While the effects of obesity and overweight are causally related to the occurrence of numerous diseases there appears to be far less known about the effects of overweight and obesity on the immune response and persistence of antibodies post-vaccination against hepatitis B virus (HBV), measles, mumps and rubella (MMR). This study sought to address this lack of data by investigating the relationship between current body mass index (BMI) in young adults and the persistence of antibodies in young adults that received HBV and MMR vaccination during childhood.
For a study of this type the inclusion criteria and subject numbers are clearly of critical importance. The authors appear to have carefully chosen the study subjects with adequate attention being paid to the timing of their vaccine doses of HBV and MMR. This is adequately described in 2.1 Inclusion criteria.
The study cohort of 2185 students were recruited consisting of 815 males and 1,370 females achieving a comparable standard of education between 2010 and 2020. The data presented in Table 1 effectively summarize the relevant details of the study population including details of BMI and overweight and underweight %.
Antibody measurements were carried out using commercially available kits.
Statistical analysis of data was carried out using valid statistical techniques and these are briefly described in section 2.4.
In a brief but adequate discussion the authors critically evaluate the results of this study by considering previous relevant and related studies. The authors point out that the sharply defined inclusion criteria included in their study enabled them to reliably conclude that there was no significant relationship between antibody persistence after HBV vaccine and BMI. This conclusion appears to be valid. Furthermore, in a new finding, the absence of a relationship was demonstrated between antibody persistence after MMR vaccine and BMI. Many readers might have the impression that BMI would influence antibody production even by physically influencing the delivery of vaccines into different tissue types. In dispelling this idea the study makes a significant contribution to current knowledge.
The assumption is made that the BMI of participants did not change significantly between childhood and adulthood and this is largely based upon the significant review of Simmonds et al (2016). Overall the literature cited in the discussion is thorough and it is generally quite relevant to this paper.
Reply: Thanks for comments and appreciation.
Round 2
Reviewer 1 Report
This reviewer has no further comments on this revised manuscript. The type of this paper could be a brief report.